# Beneficial Effects of *Castanea sativa* Wood Extract on the Human Body and Possible Food and Pharmaceutical Applications

**DOI:** 10.3390/plants13070914

**Published:** 2024-03-22

**Authors:** Taja Žitek Makoter, Mojca Tancer Verboten, Ivan Mirt, Katarina Zupančić, Darija Cör Andrejč, Željko Knez, Maša Knez Marevci

**Affiliations:** 1Laboratory for Separation Processes and Product Design, Faculty of Chemistry and Chemical Engineering, University of Maribor, SI-2000 Maribor, Slovenia; taja.zitek@um.si (T.Ž.M.); darija.cor@um.si (D.C.A.); zeljko.knez@um.si (Ž.K.); 2Faculty of Medicine, University of Maribor, Taborska 8, SI-2000 Maribor, Slovenia; 3Faculty of Law, University of Maribor, SI-2000 Maribor, Slovenia; mojca.tancer@um.si; 4Tanin Sevnica, Hermanova 1, SI-8290 Sevnica, Slovenia; ivan.mirt@tanin.si (I.M.); katarina.zupancic@tanin.si (K.Z.)

**Keywords:** *Castanea sativa*, wood extract, tannins, antioxidant activity, anti-inflammatory activity, antimicrobial activity

## Abstract

The aim of this review was to investigate the potential use of *Castanea sativa* wood extract as a food supplement and to evaluate its beneficial properties for human health. The results of the limited amount of studies suggest promising properties, including potential anti-inflammatory effects. The literature indicates that the extract, which is rich in bioactive compounds such as tannins, offers promising therapeutic possibilities for the treatment of conditions associated with chronic inflammation. Consequently, interest in its use in food and pharmaceuticals is growing. Phytochemical studies have reported antioxidant and antimicrobial activities, and anti-inflammatory, anticancer, hypolipidemic, hypoglycemic, and neuroprotective activities. A suitable extraction method and solvent is crucial for the isolation of bioactive compounds, being green extraction technologies outstanding for the industrial recovery of chestnut wood’s bioactive compounds. Nevertheless, it is important to emphasize the importance of adhering to regulatory guidelines and obtaining the necessary approvals from regulatory authorities to ensure product safety and compliance. The regulation of herbal medicinal products with proven efficacy and traditional herbal medicinal products is well defined, monitored by authorized bodies, and subject to strict control measures. It is noteworthy that medicinal products are subject to stringent quality testing to ensure safety and efficacy in use, whereas there are no comparable regulatory standards and specific labeling requirements for dietary supplements. When using herbal products, compliance with established standards in health research is essential.

## 1. Introduction

In vitro and/or in vivo assays are applied to evaluate the pharmacological activity of various compounds and their mixtures. These assays include in vitro binding assays to demonstrate the affinity of the biopharmaceutical for the target and in vivo studies to establish the potential biological activity in appropriate animal models. These so-called phytomedicines are, for all intents and purposes, plant-based drugs since they are aimed at carrying out a therapeutic action. Their effect depends on the nature and concentration of the pharmacologically active chemical constituents [1]. In modern medicine, professionals most often use pharmacotherapy to treat and strengthen their patients, while phytotherapy is often forgotten and excluded. Doctors, who have a wide range of pharmaceutical products at their disposal, are no longer interested in phytotherapy. Phytomedicinal preparations can establish homeostasis in the body and more and more scientists consider them essential as a supportive therapy in treatments [2]. One of the most interesting natural materials is tree bark [1]. Wood extracts are important natural sources of biologically active components such as polyphenols and terpenes [2], the use of which could be one of the greatest (recent) innovations in the food and pharmaceutical industries due to their numerous health-promoting properties [3,4]. These compounds have antioxidant properties and can help protect cells from oxidative damage caused by free radicals [5]. Lignans are a type of polyphenol found in wood extracts and have been studied for their potential hormone-balancing effects, particularly their estrogenic activity. Some lignans can bind to estrogen receptors and may have estrogenic or antiestrogenic effects in the body. Wood extracts also contain various terpenes and aromatic compounds found in many plants. Terpenes have been studied for their antimicrobial, anti-inflammatory, and antioxidant properties [6]. They may also contribute to the distinctive scent and flavor of wood extracts.

The combined action of extract components may play a vital role in their effectiveness [5,7]. With no defined active principles identified, it is often speculated that a range of bioactive elements could be responsible for the observed therapeutic effects, potentially acting synergistically. However, there is currently no research supporting this theory in the context of natural remedies. Consequently, the hypothesis of synergy serves as a rationale for natural products lacking identified active principles [6,8]. *Castanea sativa* water wood extract (CSWE) is rich in phytochemicals extracted from the wood of the *Castanea sativa* tree, also known as the chestnut tree. This extract is rich in various nutrients and antioxidants that can be beneficial to the human body. *Castanea sativa* wood extract is versatile and valuable in various personal care products.

CSWE has been assessed as a prospective alternative to antibiotics in animal feed [9]. In animal husbandry, particularly in intensive animal farming, numerous gastrointestinal diseases contribute to significant losses among young animals. Addressing present medical and economic challenges in intensive animal farming calls for solutions that safeguard against diseases in a safe and environmentally friendly manner.

The important compounds in CSWE are hydrolyzable tannins, which are secondary plant metabolites that precipitate proteins, inhibit digestive enzymes, and impair the utilization of vitamins and minerals, leading to chronic or subacute effects [8]. It is now known that due to the broad structural diversity of tannins (hydrolyzable/condensed/complex tannins; monomer/polymer structure), different effects of different types and amounts of tannins on humans and different animal species can be expected. Certain tannins are suitable for feeding monogastric animals and are commonly used as anthelmintics, antimicrobials, antivirals, and to aid in treating diarrheal diseases [10,11]. In addition, the structure of ellagitannins, the main hydrolyzable tannins in CSWE, indicates their reducing and antioxidant capacity, which has been demonstrated in some in vitro experiments with their representatives as geraniin, ellagic acid, and punicalagin [12,13]. However, their systemic antioxidant potential in vivo highly depends on their absorption and, to some extent, their bioavailability.

Many tannin molecules have also been shown to reduce the mutagenic effects of numerous mutagens. Many carcinogens produce oxygen-free radicals that interact with cellular macromolecules [14]. The anticarcinogenic and antimutagenic potential of tannins may be related to their antioxidant properties, which are essential for protection against cellular oxidative damage, including lipid peroxidation [15]. Both tannins and related compounds have been reported to inhibit the formation of superoxide radicals. The antimicrobial effect of tannins has been well documented [16]. Tannins have inhibited the growth of many fungi, yeasts, bacteria, and viruses. Tannic acid and propyl gallate, but not gallic acid, have also been found to inhibit food-borne bacteria, water-borne bacteria, and off-flavor microorganisms. The antimicrobial property of tannic acid can also be used in food processing to extend the shelf life of certain foods, such as meat, cheese, and fish. Tannins have also been reported to have other physiological effects, such as promoting blood clotting, lowering blood pressure, lowering serum lipid levels, inducing liver necrosis, and influencing immune responses [17,18,19,20]. The dose and type of tannins are decisive for these effects. The choice of the appropriate extraction method and solvent is crucial to ensure the purity of the substances and to allow the isolation of specific compounds without interference from other components [21]. This is crucial for the preparation of purified samples that are required for precise analytical procedures such as chromatography and spectroscopy. In the field of biological and chemical research, careful selection of extraction methods and solvents is essential to obtain pure compounds. Purity is an essential prerequisite for an in-depth study of the properties, reactions, and potential applications of these compounds in various processes. The recovery of extracted tannins depends on the properties of the raw material, such as the geographical location, biological origin, wood species, age, and location of the tree. However, their heterogeneous nature poses a challenge during extraction, mainly due to impurities such as minerals and sugars [22,23]. Another important factor is the extraction method, which affects the content of impurities. Common solvents include hot water, acetone [23,24,25], ethyl acetate [26], methanol [27,28], ethanol, sodium sulfite [29], and NaOH [30]. Parameters such as temperature, solvent ratio, and particle size have a significant influence on the extraction process and subsequent recovery rate. Enzymes or additives such as acids help to improve the quality of the tannins. Hot water extraction is still popular, as it is simple and cost-effective and yields both condensed and hydrolyzable tannins [31,32,33,34]. Extraction temperature varies depending on the solvent and raw material chosen, while particle size is proving to be a critical factor, with smaller sizes allowing faster extraction [22,27,35].

The objective of this review is to outline the phytochemical composition and health-promoting properties of CSWE, with a focus on the sustainable extraction methods utilized to recover bioactive compounds and their potential uses in the food and nutraceutical sectors.

## 2. Biological Activities

The diverse array of compounds present in CSWE arises from various linkage possibilities between esters and glucose. Multiple studies suggest that ellagitannins possess antiatherogenic, antithrombotic, anti-inflammatory, and antiangiogenic properties [36], as well as antioxidant activity [37]. Additionally, the principal ellagitannin metabolites, urolithins, demonstrate potent antioxidant activity [38]. The biological effects of tannins, such as antioxidant activity, free radical scavenging ability, antigenotoxic and antimicrobial activity, metabolic properties, etc. [39,40,41], play an essential role in using extracts in the food, cosmetic, and pharmaceutical industries [5]. Tannins are known for their astringent taste and are commonly found in foods such as fruits and specific plant parts (e.g., bark and leaves) [42]. Tannins have also been used for centuries in traditional medicine for their potential health benefits. Tannins are classified into two main categories: hydrolyzable tannins and condensed tannins [43]. Tannins with bark are usually hydrolyzable tannins. These tannins are esters of gallic acid (Figure 1) or ellagic acid (Figure 2) with sugars. When they are hydrolyzed, they break down into their respective acids and sugars. They have shown various biological activities, including antioxidant, anti-inflammatory, antimicrobial, and anticancer properties [16,44,45].

### 2.1. Antioxidant Activity

The antioxidant properties of CSWE help protect the body from free radical damage. These are unstable molecules that can damage cells and contribute to the development of chronic diseases such as cancer, heart disease, and Alzheimer’s disease [46,47]. Renault et al. evaluated the antioxidant activity of CSWE polysaccharides using the DPPH scavenging method. They found that 4-O-methylglucuronoxylans (MGX) 2 have a significant scavenging effect on DPPH radicals. At a concentration of 500 g/mL, MGX 2 eliminated DPPH radicals, and its IC50 value was determined to be 225 g/mL, suggesting its potential as an antioxidant. However, the antioxidant potential of MGX 2 was less than one-tenth that of vitamin E, which is considered a benchmark antioxidant and has an IC50 value of 25 g/mL. Their study suggests that chestnut-derived polysaccharide derivatives could serve as natural and safe alternatives to synthetic antioxidants in the food and cosmetic industries due to their potential safety and cost-effectiveness [48].

Nevertheless, further safety evaluation and research are needed to confirm their safety and efficacy as antioxidant additives [48]. A study of pigs found that a combination of oregano essential oil and sweet chestnut wood extract can prevent lipid oxidation by increasing the antioxidant status of the pig, thus extending the shelf life of the meat [42]. CSWE reduces oxidative stress biomarkers and prevents DNA damage [49]. CSWE has been shown to have potential benefits for cardiovascular health. Studies suggest that it may help lower blood pressure and cholesterol levels and improve blood flow [46]. A study by Frankič et al. showed that CSWE has in vivo antioxidant activity against oxidative stress induced by high n-3 polyunsaturated fatty acids (n-3 PUFA) in young pigs. CSWE is composed of 73% tannic agents, 18% simple sugars, 2% inorganic matter, and 7% water, where ellagitannins are the major group of hydrolyzable tannins (of which castalagin (40.5%) (Figure 3) and vescalagin (26.3%) (Figure 4) are the primary representatives). The amount of CSWE used was 3 g per kg of feed. CSWE has been shown to reduce the formation of toxic PUFA oxidation products and protect blood lymphocytes from DNA damage. The addition of 3 g CSWE per kg of feed was comparable to the effect of 90.4 mg per kg of vitamin E added [50]. A research study by Ranucci et al. showed that the use of oregano essential oil and CSWE not only increased the antioxidant potential of the serum, but also reduced lipid oxidation in pork. This combination resulted in a significantly darker and redder color of the cooked meat. In addition, the plant extract mixture significantly improved the sensory properties of the pork [42]. Cardullo et al. conducted a study on CSWE using fractionation by XAD-16. Remarkably, fraction X3 showed significant antioxidant and α-glucosidase inhibitory activities. In addition, fractions S4-S6 exhibited antioxidant properties, with fraction S7 standing out for its highest hypoglycemic activity [51]. In a research article written by Carlo Genovese et al. it was found that the ethanolic extract of *C. sativa* had the highest total content of polyphenols and flavonoids. In addition, the extract showed the strongest antioxidant activity in a dose-dependent manner, underlining its potential as a promising source of potent antioxidants [52]. Campo et al. used the Folin-Ciocalteu test with results of 0.067 and 56.99 g/100 g extract weight, the ORAC test with results of 450.4 and 3050 μmol/g Trolox equivalents/extract weight, and EC50 measured with the DPPH test with values of 0.444 and 2.399 μM. These results indicated a novel, environmentally friendly, and economically sustainable method for the extraction of chestnut fractions, which led to a chemically differentiated, stable, and reproducible composition [33]. Studies by Renault et al. showed that the IC50 of MGX extracted with water was less than 225 μg mL^−1^. In contrast, MGX extracted with alkalis showed no detectable radical scavenging. Characterization of the extracts by colorimetric assay, GC, LC–MS, and NMR spectroscopy provided valuable insights into the structure–function relationships of MGX, particularly with respect to its antioxidant activity [48]. The study conducted by Chiarini et al. showed significant effects of CSWE on cultured cardiomyocytes. The extract showed a dose-dependent reduction in the intracellular formation of reactive oxygen species and improved cell viability after oxidative stress. In addition, the extract reduced noradrenaline-induced (1 μM) contraction in guinea pig aortic strips and induced transient negative chronotropic and positive inotropic effects in guinea pig atria. Importantly, these effects were obtained independently of cholinergic or adrenergic receptors. These results suggest that CSWE has antioxidant activity and may exert cardioprotective effects [53]. Table 1 represents biological compounds with antioxidant activity in CSWE.

### 2.2. Antimicrobial Activity

The few associations of food–tannin compounds already known envisage the use of condensed tannins only. Papa et al. studied the wood extract of *Castanea sativa* against strains of *Chlamydias*. *C. Trachomatis* infects humans and is sexually transmitted. It can also infect newborns with neonatal conjunctivitis and pneumonitis. There have been reports of the failure of antibiotic treatment in humans, so there has been increasing interest in the biological activity of CSWE. In this study, the potential of a natural extract of chestnut wood (CSWE) containing hydrolyzable tannins was demonstrated as an antimicrobial agent. The compound demonstrated significant effectiveness in inhibiting the growth of various strains of *Chlamydia*, belonging to six different species. If future studies validate the in vivo effectiveness of CSWE against chlamydiae and establish its safety profile without any adverse effects, this extract has the potential to be regarded as a promising candidate for treating chronic and persistent chlamydial infections [19].

Živković et al. found that CSWE exhibited antimicrobial activity against several pathogenic bacteria, including *Escherichia coli*, *Staphylococcus aureus*, and *Pseudomonas aeruginosa*. The study concluded that the extract could potentially be used as a natural alternative to synthetic antimicrobial agents [54]. Hao et al. found that CSWE exhibited potent antimicrobial activity against several foodborne pathogens, including *Salmonella enterica and Listeria monocytogenes* [55]. Their study suggests that the extract could be used as a natural preservative in food. The studies conducted by Prapaiwong et al. showed promising in vitro results regarding the antimicrobial activity of the hydrolyzable tannin extract against subclinical mastitis bacteria. The hydrolyzable tannin extract showed comparable efficacy to the antibiotics used as positive controls, especially at concentrations of more than 630 mg/mL [56]. Zaikina et al. carried out a study which showed that the addition of herbal feed additives from chestnut wood extract as a substitute for an antibacterial drug in the feed of broiler chickens had no negative effects on the biochemical blood parameters. Instead, there was an improvement in the intensity of the nutrient digestion process [57,58]. Consequently, this addition led to improved quality indicators in the carcasses of the broilers throughout the growing period [59]. In the study conducted by Lombardi et al., the experimental results showed that CSWE exhibited remarkable growth inhibition against *Trichophyton interdigitale*. In addition, CSWE showed high activity against *Alternaria* sp. and *Rhizopus stolonifer* [60]. In a study by Lupini et al., several compounds showed extracellular antiviral activity against both avian reovirus and avian metapneumovirus, with IC50 values ranging from 25 μg/mL to 66 μg/mL. In particular, the quebracho extract showed recognizable intracellular antiviral activity against avian reovirus with an IC50 value of 24 μg/mL. These initial results indicated that the investigated plant extracts are promising candidates for combating specific avian virus infections [61]. In a study by Salami et. al, CSWE was shown to have specific antimicrobial activity against methanogens and protozoa in a long-term feeding trial without impairing rumen fermentation [62]. The results of Bahuaud et al. suggest that extracts from tannin-containing plants have the potential to influence a crucial process during the initial stages of larval invasion into the host. In particular, the introduction of PEG (polyethylene glycol) resulted in a complete or partial restoration of levels towards control, indicating the significant role of tannins in the inhibitory process. However, the study also suggests that other secondary metabolites could potentially interfere with the process and contribute to the observed differences in response between the two nematode species [63]. The experimental results of the study conducted by Romani et al. showed a consistent inhibition of mycelial growth rate in the presence of tannins for almost all fungi tested. The lowest inhibition was observed for *B. cinerea* (between 7.5% and 28.9%) and *P. italicum* (53.8% in a substrate containing 5.0% *w*/*v* sweet chestnut extract). In particular, for *F. oxysporum f.* sp. *radicis-lycopersici* and *F. solani*, the inhibitory effect was proportional to the tannin concentration and ranged between 33.7% and 56.6%, while, for *R. solani*, it ranged between 29.7% and 68.8% and, for *P. digitatum*, between 64.7% and 87.0%. The most pronounced inhibitory effect was observed in *S. rolfsii*, with a range of 5.0% to 100% [64]. Overall, these studies suggest that the wood extract of *Castanea sativa* has promising antimicrobial properties and could potentially be used in various applications in the food, pharmaceutical, and cosmetic industries. However, further research is needed to fully understand this compound’s potential benefits and limitations. Table 2 represents biological compounds with antimicrobial activity in CSWE.

### 2.3. Anticancer Activity

There is evidence in the literature that CSWE may have potential anticancer properties, including the ability to inhibit cancer cell growth and induce apoptosis (programmed cell death) in cancer cells. However, further research is needed to understand these potential benefits entirely [11]. Chemoprevention aims to prevent or halt cancer development, and natural extracts and herbal medicines are of interest because of their phytochemical composition. Lenzi et al. studied the effect of CSWE on human T-leukemia cells and found that it can induce apoptosis in a dose- and time-dependent manner without affecting the cell cycle. However, at high concentrations and prolonged treatment, it also triggered necrosis. CSWE may be a partially selective cytotoxic agent, as it induced higher cell death in transformed cells than in nontransformed cells. Further studies are needed to fully understand its potential as a chemopreventive agent [42]. The studies conducted by Frédérich et al. showed a strong growth inhibitory activity in vitro against human LoVo colon cancer. These results suggest that the compounds investigated have the potential to serve as readily available sources of new anticancer agents with significant therapeutic effects [65]. In a study conducted by Pérez et al., two triterpenoids were isolated and tested for their cytotoxicity against two cancer cell lines (PC3 and MCF-7) and normal lymphocytes. Their results showed a stronger effect on breast cancer cells (MCF-7) compared to prostate cancer cells (PC3). In particular, chestnoside B (2) showed the highest cytotoxicity with an IC50 of 12.3 μM against MCF-7 cells, outperforming the positive controls. However, it showed moderate activity against normal lymphocytes (IC50 = 67.2 μM). These results emphasize the presence of triterpenoid saponins in the hearts of sweet chestnut and their potential for breast cancer chemoprevention [66]. In a preliminary investigation by Sorice et al., the influence of a hydroalcoholic extract derived from chestnut shells was examined on the normal human keratinocyte HaCaT cell line, along with five different human tumor cell lines: A375 (malignant melanoma), H460 (lung cancer), HepG2 (liver hepatocellular carcinoma), HT29 (colorectal adenocarcinoma), and MCF7 (estrogen-receptor-positive breast cancer). Metabolomic profiling using 1H-NMR analysis revealed alterations in amino acid levels and other metabolites upon treatment with polyphenols extracted from chestnut shells [67]. These findings underscore the impact of biomolecules on various cellular processes, including proliferation, apoptosis, cell cycle regulation, and mitochondrial function, as well as cytokine expression and metabolite profiles. Additionally, the study discussed the predominant phenolic compounds in chestnut peel extracts, with gallic acid being the most abundant, followed by ellagic acid and syringic acid, while quercetin and rutin were present in lower concentrations. Table 3 represents biological compounds with anticancer activity in CSWE.

### 2.4. Anti-Inflammatory Activity

CSWE has proven anti-inflammatory properties that can help reduce inflammation in the body. Chronic inflammation has been linked to many health problems, including arthritis, diabetes, and heart disease [46]. Papa et al. demonstrated that CSWE reduced all *Chlamydia* strains tested at 1 µg/mL, while SMAP-29 at 10 µg/mL caused a reduction in the infectivity of *C. trachomatis* and *C. pneumoniae*. A strong reduction in the infectivity of *C. trachomatis*, *C. pneumoniae*, and *C. abortus* was achieved with a CSWE concentration of 10 µg/mL, while their infectivity was almost inhibited at a CSWE concentration of 100 µg/mL [68]. The study by Brizi et al. investigated the neuroprotective effect of CSWE in in vitro models of oxidative stress injury. Their results suggest that CSWE could be a valuable support as a dietary supplement, as it combines positive preventive neuroprotective effects with high antioxidant activity [69]. Budriesi et al. studied the protective effects of CSWE. CSWE exhibited consistent effects in restoring antioxidant enzymatic activity and rebalancing pro-inflammatory and anti-inflammatory cytokines in the livers of rats on a high-fat diet. It decreased TNF-α, a key factor in Non Alcoholic Fatty Liver Disease (NAFLD) and Non Alcoholic Steato Hepatitis (NASH) development, as well as IL-6, IL-10, and IL-17A, which are implicated in obesity and NAFLD regulation. ENC1 shows promise as a therapeutic compound for treating NAFLD and NASH by reducing oxidative stress and modulating cytokine levels [50,70]. Methanol extract from the bark of 14 different tree species has also been shown to be important for anti-inflammatory activity in relation to antiprotease activity (trypsin, thrombin, and urokinase) using chromogenic bioassays. Knez et al. explored the trypsin protease inhibitory potential with IC50 values below 10 microg/mL [71]. The most significant inhibitory effects were observed in methanol extracts of Acer platanoides (IC50 = 1.8 microg/mL), Rhus typhina (IC50 = 1.2 microg/mL), and Tamarix gallica (IC50 = 1.7 microg/mL). However, the outcomes differed when tested against thrombin compared to trypsin. Castanea sativa (IC50 = 73.2 microg/mL), Larix decidua (IC50 = 96.9 microg/mL), and Rhus typhina (IC50 = 20.5 microg/mL) exhibited the most notable inhibitory effects on thrombin. The study suggests that these extracts hold promise as natural sources of serine protease inhibitors. In another study, research by Orso et al. [72] explored the potential anti-inflammatory and antioxidant properties of hydroalcoholic (H_2_O:EtOH 50:50) and aqueous extracts of *C. sativa* woods. The analysis included individual ellagitannins and total phenolic compounds, followed by assays using an in vitro model of undifferentiated human intestinal Caco-2 cells (colonocytes) stimulated with IL-1β-IFNγ. The experiments conducted on male grown zebrafish revealed that tannins treatment improved the microbiota composition significantly, as confirmed by histopathological analysis. Although the morphology of the tissue appeared restored, pro-inflammatory cytokines TNFα, IL-8, and IL-1β remained elevated, while COX-2 decreased [73]. Table 4 represents biological compounds with anti-inflammatory activity in CSWE.

## 3. Food and Pharmaceutical Applications

### 3.1. Food Supplement

The use of *Castanea sativa* wood extract as a food supplement is an area that requires further scientific investigation and regulatory considerations. Before *Castanea sativa* wood extract is used as a food supplement, its safety must be assured and regulatory requirements must be met. Regulatory agencies vary by country or region and have specific guidelines for the use of dietary supplements. It is critical to consult with the appropriate authorities to ensure compliance with local regulations. Ensuring quality and standardization is also an important step. If you are considering *Castanea sativa* wood extract as a food supplement, choose products from reputable manufacturers that perform quality control checks. Standardizing the extract’s composition, including bioactive compound content, is critical for consistency and efficacy. Researching the specific health benefits, optimal dosage, and safety profile is also essential [32]. Many studies have already been conducted on animals showing the potential of CSWE extract as a dietary supplement. For example, Minieri et al. carried out an experiment with 80 laying hens from two autochthonous Tuscan breeds: 40 from the Mugellese breed (MU) and 40 from the White Leghorn breed (WL). The experimental groups were fed a diet supplemented with 2 g of *Castanea sativa* Mill. bark extract (CSWE). Then, 70 eggs were randomly sampled and analyzed for cholesterol content, fatty acid (FA) profile, weight, shell thickness, and yolk color. Physical parameters such as yolk color and egg quality indices were not affected by the treatments. However, the concentration of unsaturated fatty acids increased, while the cholesterol content decreased significantly, by 17% in WL and 9% in MU. Supplementation with CSWE resulted in a shift in lipid composition, which favored a healthier egg quality [74]. The study on the neuroprotective effects of *Castanea sativa* suggests that the natural extract of CSWE has the potential to serve as a useful dietary supplement due to its neuroprotective effects and high antioxidant potential. By protecting human oxygen species (ROS), CSWE could help prevent neuronal diseases [75]. Furthermore, by reducing DNA damage caused by stress, CSWE may also help prevent cell death resulting from apoptosis. These findings indicate that CSWE could be a beneficial addition to a neuroprotective diet, potentially helping prevent oxidative stress-related brain damage [69]. In a study conducted by Ranucci, CSWE and oregano extract were administered to pigs, which resulted in a significant reduction in lipid oxidation in the meat [42]. In the study conducted by Mergedus et al., the results obtained in fattening bulls showed that the addition of hydrolyzable tannins (possibly a substance or treatment) is both practical and justified. The addition of hydrolyzable tannins showed an improvement in growth performance and feed efficiency, with no adverse effects on carcass and meat quality observed [76]. In a study conducted by Ranucci et al., an improvement in the oxidative stability of pork was observed with the addition of CSWE, and this improvement was also highlighted in the preservation of color [42]. Several meat processing techniques involve the integration of natural extracts to enhance the nutritional profile and health advantages of meat, alongside the utilization of mixed or refined natural antioxidants to ensure the chemical safety of cooked and processed meats [77]. Schiavone et al. [78] conducted a study on broiler chickens fed with 0.20% chestnut wood extracts, observing improved growth performance without significant effects on digestibility, carcass characteristics, or nitrogen balance. Conversely, Voljč et al. [79] found no impact on in vivo and in vitro oxidative stress markers or meat lipid stability when broiler chickens were fed with 3 g of chestnut wood extract/kg diet. Another method involves incorporating chestnut shell extract into cheese to enhance its nutritional value and extend its shelf life, thus reducing food industry waste. Due to the presence of catechin and gallic acid, chestnut shell extract displays notable antioxidant properties. Ferreira et al.’s study demonstrated that the incorporation of phenolic extract did not affect the cheese’s pH value, maintaining it within the range of 6.2–6.5. Moreover, the extract reduced syneresis, enhancing stability. Higher concentrations of the extract were found to inhibit a greater percentage of free radicals. While incorporating chestnut shell extract into fresh cheese shows promise for adding value to dairy products and promoting sustainable food production, it may affect the organoleptic properties of the product. Thus, while this approach presents an innovative perspective for utilizing agro-industrial by-products to extend the shelf life of fresh cheese, further research is needed to optimize its sensory attributes and consumer acceptance [80].

### 3.2. Impact on the Gastrointestinal System

CSWE may have potential benefits for digestive health, as suggested by some studies indicating its potential to reduce symptoms of irritable bowel syndrome (IBS) and improve overall gut health [81]. Stop Fitan^®^, a bioactive purified natural extract of chestnut wood (*Castanea sativa*), has been studied for its effects on guinea pig ileum and proximal colon tissues by Budriesi et al. Their results indicate that CSWE has a spasmolytic effect on both the ileum and the proximal part of the colon, suggesting its potential suitability for the treatment of diarrhea [70]. CSWE is also crucial for the extraction of plant tannins, with the extract containing approximately 75% of active tannins. The main constituent, castalagin, along with smaller amounts of vescalagin, castalin, and vescalin, has shown promise in the prevention and treatment of diarrhea in pigs and cattle caused by dietary changes. This positive effect is attributed to castalagin’s ability to prevent water loss via mucous membranes and its potential to form chelates with iron, influencing the absorption of the metal in the animal’s digestive tract [82]. Moreover, CSWE has been associated with beneficial effects on cardiovascular parameters and has been shown to counteract weight gain [83]. In the study conducted by Mattioli et al., the effect of CSWE with different chemical compositions on the spontaneous contractility of isolated intestinal tissue from healthy chickens was investigated. Their results showed that the chemical composition of phytocomplexes has different effects on spontaneous intestinal contractility and influences both the tone and the further course of the food bolus [43]. In the study conducted by Micucci et al., it was observed that CSWE increases the contraction of the gallbladder and causes relaxation of the sphincter of Oddi. These effects suggest a potential benefit in pathological conditions characterized by prolonged transit time, especially in individuals at risk of gallstones [83]. In the study conducted by Brus et al., it was observed that hydrolyzable tannins (HT) from the wood extract Tanex at a concentration of 4 µg/mL upregulate the expression of GLUT2 and increase glucose uptake at 1 µg/mL. In addition, the time-dependent passage of gallic acid through enterocytes was affected by all wood extracts compared to gallic acid itself as a control. These results suggest that hydrolyzable tannins can modulate glucose uptake and the passage of bile acid in the 3D cell model used in the study [84].

### 3.3. Impact on Skin

*Castanea sativa* wood extract has been studied for its potential benefits for skin health [85]. It has been shown to have antioxidant and anti-inflammatory properties, which may help to protect the skin against damage and reduce the appearance of wrinkles and other signs of aging. Vayalil et al. investigated the effects of *Castanea sativa* wood extract on skin health, specifically its ability to inhibit matrix metalloproteinases (enzymes that degrade collagen and elastin) and enhance type I procollagen synthesis in human skin fibroblasts. Their results suggest that the extract has potential as a natural antiaging ingredient in cosmetics [86].

### 3.4. Impact on Cardiovascular System

Ellagitannins may have a beneficial effect on the cardiovascular system. In the studies by Micucci et al. and Bevc et al., rats were fed a high-fat diet supplemented with an extract of *Castanea sativa* wood. A high-fat diet leads to what is known as metabolic syndrome. This type of diet can lead to insulin resistance and high blood sugar, high blood cholesterol and triglyceride levels, hypertension, and cardiovascular disease. The extract used in the study contained the hydrolyzable tannins castalin, vescalin, castalgin, vescalgin, ellagic acid, and gallic acid. Briefly, the effects on body weight, serum biochemical parameters, and inflammatory cytokines were studied. It was found that the addition of CSWE to a high-fat diet reduced weight gain and serum lipids. The results of the studies indicated that ENC can restore metabolic dysfunction and cholinergic muscarinic receptor function in the heart of rats, suggesting that CSWE is a potential dietary supplement for the treatment of obesity [83,87].

Chiarini et al. investigated the cardioprotective effects of *Castanea sativa* wood extract (CSWE). This study was conducted using primary cultures of neonatal rat cardiomyocytes to investigate the antioxidant and cytoprotective effects of CSWE. Guinea pig left and right forearms, left papillary muscle, and aorta were then isolated to evaluate the effects on cholinergic and adrenergic responses. Bark extract was found to have important antioxidant and cytoprotective effects and to modulate some important cardiac functions. Their results clearly indicate the nutraceutical value of CSWE bark extract and therefore suggest the potential for its use as a dietary supplement for the prevention of cardiovascular disease [53]. Santulli et al. investigated the neuroprotective effects of CSWE in human U-373 MG astrocytoma cells and rat cortical slices under ischemic conditions [88]. CSWE showed a neuroprotective effect when administered at certain concentrations before injury, while its addition during reperfusion or after injury showed no effect. These results suggest the potential of CSWE as a preventive neuroprotective agent [88]. Budriesi et al. investigated the effects of CSWE in overweight rats on a high-fat diet (HFD). They studied CSWE in 120 male Sprague–Dawley rats. Over a 21-day period, weight gain, serum lipids, plasma cytokines, liver histology, microsomal enzymes and oxidation, intestinal oxidative stress, and contractility were examined. CSWE increased body weight, pro-inflammatory cytokines, microvesicular steatosis of hepatocytes, altered microsomal enzymes, and induced oxidative stress in the liver and intestine, which impaired intestinal contractility. CSWE showed antiobesity and antioxidant effects in CSWE-fed rats, suggesting a potential approach for the treatment of obesity-related diseases [57].

## 4. Legislative Regulation for the Use of *Castanea sativa* Wood Extract Products, Both in Pharmaceutical Application and Food

If products made from *Castanea sativa* wood extract were to be placed on the market or for consumer purposes, such products should be classified in one of the legislative groups of products for which they should fully meet the conditions. Medicinal plant products in pharmaceutical forms belong to one of the following groups: medicines, nutritional supplements, cosmetic products, or medical devices [53].

Considering the effects of *Castanea sativa* wood extract, only two groups are highlighted below in the presentation of the legislative regulation for use, namely, medicines and nutritional supplements. In accordance with Article 1 of Directive 2001/83/EC of 6 November 2001 on the Community code on medicinal products for human use (Directive 2001/83/EC) [53], a medicinal product is defined as any substance or combination of substances presented with properties for the treatment or prevention of disease in humans; or that can be used in or administered to humans to restore, improve, or modify physiological functions through pharmacological, immunological, or metabolic action, or to establish a diagnosis. On the basis of point 3 of article 5 of the Medicinal Products Act (ZZrd-2) [53], the substance can be of plant origin. Dietary supplements are defined in Directive 2002/46/EC of the European Parliament and of the Council of 10 June 2002 on the approximation of the laws of the Member States on dietary supplements and the Regulations on dietary supplements (Directive 2002/46/EC) [53]. They are categorized as foods intended to complement a regular diet, serving as concentrated sources of individual or combined nutrients or other substances with nutritional or physiological effects. These products are available in various forms, such as capsules, lozenges, and tablets, as well as powders in bags, liquid ampoules, dropper bottles, and similar liquid and powder forms meant to be consumed in measured, small unit quantities.

### 4.1. Registration of Products from Castanea sativa Wood Extract as Pharmaceutical Drugs

Directive 2001/83/EC was adopted within the framework of the European Union as a basis for the actions of the members in the procedures for the registration of traditional medicinal products of plant origin, which includes *Castanea sativa* wood extracts in human medicine, Commission Regulation (EC) No. 1084/2003 of 3 June 2003 on the review of changes to the conditions of the marketing authorization for medicinal products for human use and medicinal products for veterinary use, issued by the competent authority of the Member State (Regulation 1084/2003/EC) [53] and Regulation (EC) No. 726/2004 of the European Parliament and of the Council of 31 March 2004 on Community procedures for obtaining marketing authorization and the control of medicinal products for human and veterinary use and on the establishment of the European Medicines Agency (Regulation 726/2004/EC) [53]. In the Republic of Slovenia, it is necessary to obtain a permit for medicinal products of plant origin and traditional medicinal products of plant origin from the Public Agency of the Republic of Slovenia for Medicines and Medical Devices (JAZMP) requirements as for other medicines. The special feature is the simplified process, which is legally defined for obtaining marketing authorization for a traditional herbal medicine, but under the conditions that it has therapeutic indications suitable exclusively for traditional herbal medicines, which, due to their composition and purpose, are suitable for self-medication; that it is intended exclusively for use in accordance with the specified strength and dosage; that it is for oral or external use or for inhalation; that the period of traditional use of the drug has expired; and that the product has been proven not to be harmful under certain conditions of use, and the pharmacological effects or effectiveness of the drug are probable based on long-term use and experience. The procedure is carried out in accordance with the *Rulebook on Traditional Medicinal Products of Plant Origin* [53].

### 4.2. Use of Castanea sativa Wood Extract as a Food Supplement

Dietary supplements, which are concentrated individual nutrients or their combinations, are classified as foodstuffs in the European Union and before use must comply with general food legislation, namely, the Health Act on the Health Suitability of Foodstuffs and Products and Substances that Come into Contact with Foodstuffs (ZZUZIS) [53]. Food supplements should meet the standard of microbiological and radiological integrity; they must not contain pesticides, heavy metals, illegal, or inappropriate additives; and they must not be mechanically contaminated with impurities that can be harmful to human health.

It should also be pointed out that no special instructions for use are used in the labeling of the nutritional supplement, nor is there a note about the need to consult a doctor or pharmacist when consuming it. In accordance with Regulation (EC) No. 1924/2006 of the European Parliament and of the Council of 20 December 2006 on nutrition and health claims on foods [53], approved nutrition and health claims are allowed if the conditions for sufficient content of a certain substance in the food are met.

## 5. Conclusions

While the wood extract of *Castanea sativa* wood extract has shown promising properties, including potential anti-inflammatory effects, further research is needed to determine its suitability for food and pharmaceutical applications. Here are some important points to consider: Potential anti-inflammatory effects: *Castanea sativa* wood extract contains bioactive compounds such as tannins that have shown anti-inflammatory effects in certain studies. These effects suggest that the extract has therapeutic potential for the treatment of conditions associated with chronic inflammation. Phytochemical composition: the wood extract of *Castanea sativa* contains a variety of bioactive compounds, including phenolic compounds such as tannins. These compounds contribute to the potential health benefits of the extract and could make it an interesting candidate for food and pharmaceutical applications. Since ancient times, *Castanea sativa* wood extract has been used by different cultures for various health purposes. Traditional knowledge and ethnopharmacological studies may provide valuable insights into the potential uses and safety of *Castanea sativa* wood extract. Although initial studies have shown promising results, further research is needed to understand the extract’s mechanisms of action, evaluate its safety profile, and determine optimal dosages for specific applications. Further in vitro animal and clinical studies are needed to confirm its efficacy and safety. When considering the use of *Castanea sativa* wood extract in food or pharmaceutical applications, it is essential to follow regulatory guidelines and obtain the appropriate approvals from regulatory agencies to ensure product safety and compliance.

Despite the needs of modern medicine to strengthen patients with phytomedicinal preparations such as *Castanea sativa* wood extract, which could be used due to its many health benefits, it is necessary to be aware of what is safer for patients. The fact is that the procedures for the regulation of medicinal products of plant origin with proven effectiveness and traditional medicinal products of plant origin are very strictly defined; they are carried out by authorized bodies and the circulation itself is also more strictly controlled. The fact is that only medicines have a verified quality, and when used, they guarantee a certain safety and effectiveness. Dietary supplements do not meet the same standards, and their labeling and use are not specifically regulated. When using individual products of plant origin, there must be health research carried out in accordance with the relevant standards, which means that it is safer for patients to use medicines and not nutritional supplements. Most tannin-related studies have been conducted on extracts instead of pure compounds, making it difficult or impossible to compare the activity of individual compounds or to assess structure–activity relationships. As mentioned above, this review discusses the biological activity of tannins in *Castanea sativa* wood extract, and also indicates a certain biological activity of separate compounds. As tannins appear in natural tissue and are often present in a mixture, their chemical structure may not be the only parameter that determines physical and chemical properties and, above all, their biological activity. To summarize, the existing research is insufficient, especially in the field of antitumor and antifungal activity of ellagitannins, to conclude which compound is responsible for a certain activity. The main biological and pharmacological effects reported for condensed tannins can be classified into antibacterial and antiviral activities, enzyme inhibition, antioxidative effects, and antimutagenic and antitumoral properties. Their estimated interaction with biological systems originates in principle directly from the physical and chemical properties of the polyphenolic skeleton, although noticeable distinctions have been detected.

More studies are needed to investigate the interaction of tannins with specific cell lines and microbial communities to obtain a complete overview of their potent and consequent applicability. In summary, while *Castanea sativa* wood extract shows promise and has been traditionally used for various health purposes, further scientific research is needed to determine its efficacy, safety, and optimal use in the food and pharmaceutical industries.

## Figures and Tables

**Figure 1 plants-13-00914-f001:**
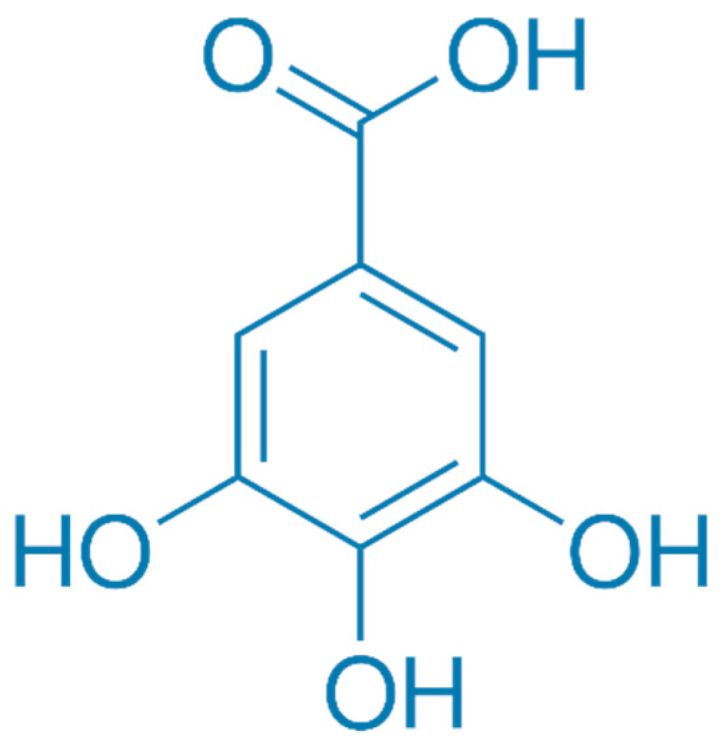
Chemical structure of gallic acid.

**Figure 2 plants-13-00914-f002:**
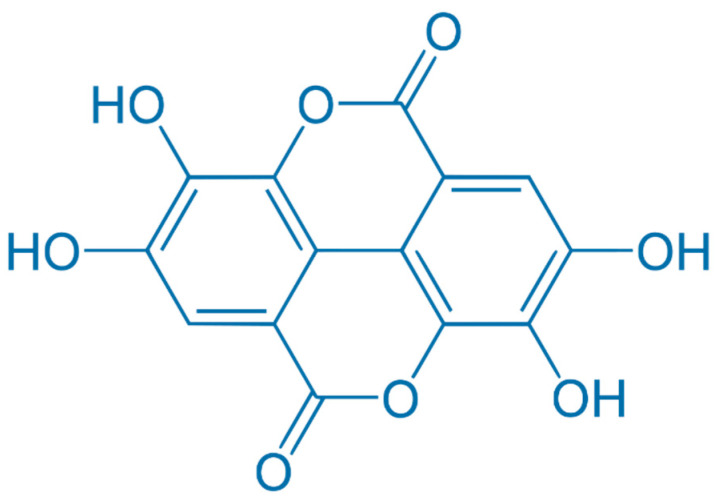
Chemical structure of ellagic acid.

**Figure 3 plants-13-00914-f003:**
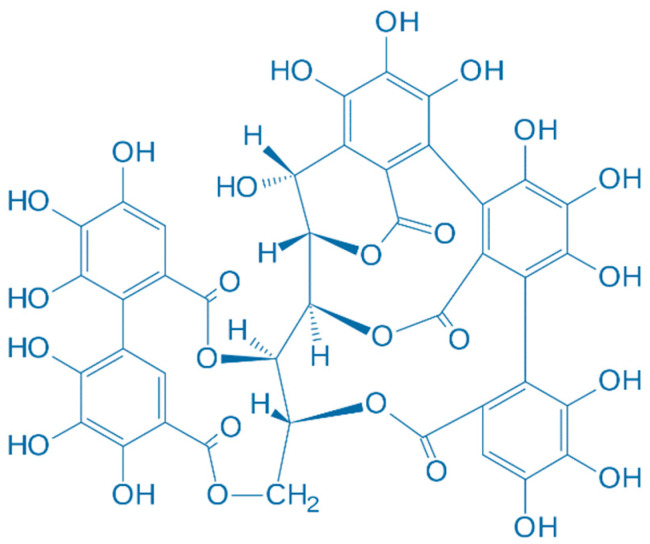
Chemical structure of castalagin.

**Figure 4 plants-13-00914-f004:**
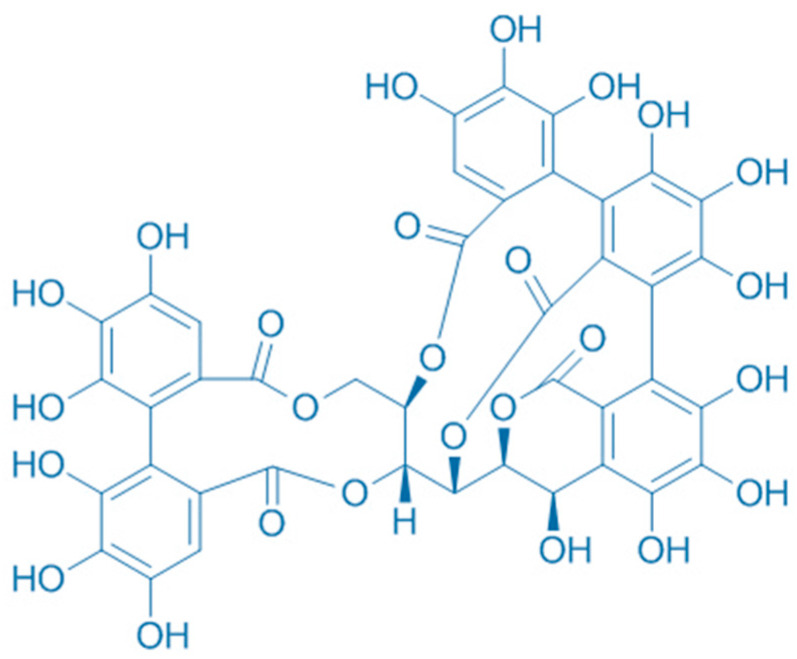
Chemical structure of vescalagin.

**Table 1 plants-13-00914-t001:** Biological compounds with antioxidant activity.

Biological Compound	Biological Activity	Reference
Derived polysaccharide derivatives (4-O-methylglucuronoxylans)	Antioxidant activity (DPPH IC50 225 μg/mL)	[48]
CSWE	Reduces oxidative stress biomarkers and prevents DNA damage (flow cytometry analyses of Jurkat cells)	[49]
CSWE	Lowers blood pressure and cholesterol levels and improves blood flow (DPPH)	[46]
Castalagin and vescalagin	Antioxidant activity (in vivo on pigs)	[50]
CSWE	Antioxidant activity, improves digestion (DPPH, ORAC)	[51]
CSWE	Antioxidant activity, antimicrobial activity, and anticancer activity (DPPH)	[52]
CSWE	Antioxidant activity (^13^C-NMR and FTIR spectrometry, with HPLC)	[7]
CSWE	Antioxidant activity, antiradical activity (DPPH)	[33]
MGX	Understanding structure–function relationships of MGX in connection with their antioxidant activity. (DPPH)	[48]
CSWE	Bark extract exhibits antioxidant activity and might induce a cardioprotective effect (primary cultures of neonatal rat cardiomyocytes)	[53]

**Table 2 plants-13-00914-t002:** Biological compounds with antimicrobial activity.

Biological Compound	Biological Activity	Reference
CSWE	Antibacterial activity against strains of *Chlamydias C. Trachomatis* infects humans and is sexually transmitted	[19]
CSWE	Several pathogenic bacteria, including *Escherichia coli*, *Staphylococcus aureus*, and *Pseudomonas aeruginosa*	[54]
Hydrolyzable tannins	Antibacterial activity (bacteria causing the mastitis)	[56]
CSWE	For feed—as an alternative to antimicrobial drugs (in vivo on Cobb-500 cross broiler chickens)	[59]
CSWE	Antimicrobial agents against pathogenic filamentous fungi (*Aspergillus brasiliensis*, *Alternaria* sp., *Rhizopus stolonifer*, and *Trichophyton interdigitale*)	[60]
CSWE	Antimicrobial activity and anticancer activity (*Escherichia coli*, *Bacillus subtilis*, *Saccharomyces cerevisiae*, and *Aspergillus niger*)	[52]
CSWE	Antiviral activity (in vitro against avian reovirus and metapneumovirus)	[61]
Hydrolyzable tannins	Positive influence on the ruminal microbiome with specific antimicrobial activity against methanogens and protozoa (in vivo study on lambs)	[62]
CSWE	Antiparasitic action (in vitro study on *Haemonchus contortus* and *Trichostrongylus colubriformis*)	[63]
CSWE	Antifungal action (in vitro inhibitory activity against *Fusarium oxysporum f.* sp. *radicis-lycopersici, Fusarium solani, Rhizoctonia solani, and Sclerotium rolfsii*)	[64]

**Table 3 plants-13-00914-t003:** Biological compounds with anticancer activity.

Biological Compound	Biological Activity	Reference
CSWE	Induce apoptosis in a dose- and time-dependent manner without affecting the cell cycle	[49]
CSWE	Inhibit cancer cell growth and induce apoptosis	[11]
CSWE	Anticancer effect (in vitro growth inhibitory activity against the human LoVo colon cancer, PC3 prostate cancer, and U373 glioblastoma cell lines)	[65]
Triterpenoids: chestnoside A in chestnoside B	Potential for the chemoprevention of breast cancer	[66]
CSWE	Antioxidant activity, antimicrobial activity, and anticancer activity (MCF-7 breast cancer cell line by the MTT assay and reactive oxygen species (ROS) determination)	[52]

**Table 4 plants-13-00914-t004:** Biological compounds with anti-inflammatory activity.

Biological Compound	Biological Activity	Reference
CSWE	Anti-inflammatory action	[46]
CSWE	Reduced all *Chlamydia* strains tested at 1 µg/mL	[68]
CSWE	Support as dietary supplement, combining beneficial preventive neuroprotective effects with a high antioxidant activity	[69]
CSWE	Anti-inflammatory action	[61]

## Data Availability

All data are collected from open source with detailed description in cited References.

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
