# Peer review of "Beneficial Effects of Castanea sativa Wood Extract on the Human Body and Possible Food and Pharmaceutical Applications"

_plants, 2024, doi:10.3390/plants13070914_

Round 1
Reviewer 1 Report
Comments and Suggestions for Authors
plants-2901652
Line 20 “not only” and “but also” not necessary restructure lines 20 and 21
Line 32 Keywords put first Castanea sativa after wood extract
Lines 55 and 56 restructure delete tannins and lignans only use polyphenols
Lines 82 to 84 the text of the topic not necessary
Line 95 include “and” as “acid and punicalagin
Lines 98 to 100, restructure the text
Line 145 include and as bark and leaves
Lines 233, 234, 246, (microorganism are italics) and 247 , 264, 266, 268,269, 340, 341, 343, 344, 383.342, 634, 640, 644.679, 684,732745, 751,753,763, 767,777,782,811,819,843,864, and 868-lower case letter (plants are italics) and line 894 change andReperfusion as and reperfusion
Please check all manuscript
Lines 296 Chestnoside change as chesttnoside
Line 379-380 check the letter format
References
631 change to lowercase
636 change in vitro, in vivo to italics
Please check pdf all is in yellow

minor editing of English language is required
Author Response
Dear Reviewer,
The authors appreciate Your time and efforts in reviewing our manuscript. We are submitting the revisions based on the remarks/comments below.
Line 20 “not only” and “but also” not necessary restructure lines 20 and 21
Line 32 Keywords put first Castanea sativa after wood extract
Lines 55 and 56 restructure delete tannins and lignans only use polyphenols
Lines 82 to 84 the text of the topic not necessary
Line 95 include “and” as “acid and punicalagin
Lines 98 to 100, restructure the text
Line 145 include and as bark and leaves
Lines 233, 234, 246, (microorganism are italics) and 247 , 264, 266, 268,269, 340, 341, 343, 344, 383.342, 634, 640, 644.679, 684,732745, 751,753,763, 767,777,782,811,819,843,864, and 868-lower case letter (plants are italics) and line 894 change andReperfusion as and reperfusion
Please check all manuscript
Lines 296 Chestnoside change as chesttnoside
Line 379-380 check the letter format
References
631 change to lowercase
636 change in vitro, in vivo to italics
Please check pdf all is in yellow
The remarks and unclarities have been addressed. Please see the revised/marked version of the manuscript.
Reviewer 2 Report
Comments and Suggestions for Authors
The various activities and legal standards of Castanea sativa wood extract are described. While the activities are particularly well-explained, with a focus on tannins, it would be beneficial to provide a more detailed discussion on the correlation between tannins and activity. It is known that various tannins exhibit activity. Please summarize the activities associated with each of these tannins.
Author Response
Dear Reviewer,
The authors appreciate Your time and efforts in reviewing our manuscript. We are submitting the revisions based on the remarks/comments below.
Remark: The various activities and legal standards of Castanea sativa wood extract are described. While the activities are particularly well-explained, with a focus on tannins, it would be beneficial to provide a more detailed discussion on the correlation between tannins and activity. It is known that various tannins exhibit activity. Please summarize the activities associated with each of these tannins.
Answer: Most of tannin-related studies have been conducted on extracts instead of pure com-pounds, making it difficult or impossible to compare the activity of individual com-pounds or to assess structure–activity relationships. As mentioned above, the review discusses the biological activity of tannins in the Castanea sativa wood extract and also indicates a certain biological activity of separate compound. As tannins appear in nat-ural tissue and are often present in a mixture, their chemical structure may not be the only parameter that determines physical and chemical properties and above all, their biological activity. To summarize, the existing research is insufficient, especially in the field of anti-tumor and antifungal activity of ellagitannins, to conclude which com-pound is responsible for a certain activity. The main biological and pharmacological effects reported for condensed tannins can be classified into antibacterial and antiviral activities, enzyme inhibition, anti-oxidative effects, antimutagenic and antitumoral properties. Their estimated interaction with biological systems originates in principle directly from the physical and chemical properties of the polyphenolic skeleton, alt-hough noticeable distinctions have been detected.
More studies are needed to investigate the interaction of tannins with specific cell lines and microbial communities to obtain a complete overview of their potent and conse-quent applicability.
Please see the revised/marked version of the manuscript. An additional paragraph has been included in the concluding section.